# A Wavelet Scattering Feature Extraction Approach for Deep Neural Network Based Indoor Fingerprinting Localization [note 1]

**DOI:** 10.3390/s19081790

**Published:** 2019-04-14

**Authors:** Bedionita Soro, Chaewoo Lee

**Affiliations:** Department of Electrical and Computer Engineering, Ajou University, Suwon 16499, Korea; sorobedio@ajou.ac.kr

**Keywords:** fingerprinting, indoor localization, indoor positioning, wavelet scattering, feature extraction

## Abstract

The performance of an Artificial Neural Network (ANN)-based algorithm is subject to the way the feature data is extracted. This is a common issue when applying the ANN to indoor fingerprinting-based localization where the signal is unstable. To date, there is not adequate feature extraction method that can significantly mitigate the influence of the receiver signal strength indicator (RSSI) variation that degrades the performance of the ANN-based indoor fingerprinting algorithm. In this work, a wavelet scattering transform is used to extract reliable features that are stable to small deformation and rotation invariant. The extracted features are used by a deep neural network (DNN) model to predict the location. The zero^th^ and the first layer of decomposition coefficients were used as features data by concatenating different scattering path coefficients. The proposed algorithm has been validated on real measurements and has achieved good performance. The experimentation results demonstrate that the proposed feature extraction method is stable to the RSSI variation.

## 1. Introduction

Indoor location-based services such as tracking, patient monitoring, navigation, and localization are relevant for today’s society and smart cities [1,2]. However, the GPS system is generally inefficient in indoor and some outdoor environments due to the signal attenuation and weakened or unavailable signal respectively. To support such services and overcome the limitations of the GPS system, several approaches have been investigated. Among them, approaches based on the receiver signal strength (RSS) measurement such as fingerprinting algorithms have attracted a lot of attention. The fingerprinting method consists of two main phases: the offline phase, and the online phase. During the offline phase, the RSSI data is collected in the area of interest at some predefined points called Reference Points (RPs) with known position coordinates. Then, the radio map is built from the collected data. The radio map contains one sample of the RSSIs of each Access point (AP) at each reference point. In the online phase, when a new RSSI data is collected in the same area, the location of that measured RSSI is estimated by finding the closest match between the current measured RSSI data and those in the radio map. The main advantage of RSS-based localization methods is that they do not require a specific network infrastructure or hardware to effectively operate. Also, these days, smartphones have built-in RSSI measurement functionalities that can accurately measure the RSSIs of most available radio networks such as Wi-Fi and Bluetooth.

However, RSSIs are very unstable and their variability degrades the performance of most existing RSS-based localization methods. In the literature, a lot of research has been done to improve RSS-based methods for outdoor environments based on different radio propagation models analysis [3]. Also, weather conditions (snowing or raining) can impact on the outdoor RSS-based localization method [4]. Due to the indoor environmental charateristics, those outdoor localization models barely work in indoor environments. Several studies have been carried-out to mitigate the effect of RSSI variation on the positioning perfermance in both indoor and outdoor environments. Those studies include methods called weighted centroid or relative-span exponential weighted localization which are accurate than a generic k-nearest neighbor (KNN) or trilateration method [5]. These methods require a wise choice of the weights.

Morever, there exist a dedicated event to indoor localization specifically International conference on Indoor Positioning and Indoor Navigation (IPIN) [6]. IPIN presents most potential accurate indoor localization algorithms including both accademic and industrial methods through different programs. In addition, generally, RSSI fingerprinting-based algorithms are the methods which achieved the best localization results at IPIN. Although the methods presented at IPIN achieved good localization performance, the dataset used during such an event is still complex to exploit specifically the floor ground labels. Therefore, few researchers have exploited this dataset in their performance evaluation.

Further approaches have been proposed to improve the localization performance through the use of Artificial Neural Network (ANN). Unfortunately, such methods generally perform below par due to the RSSI’s fluctuation. In fact, an ANN-based fingerprinting method requires a good feature extraction method to achieve a good result. To improve the robustness of the features, several approaches have been discussed such as approaches combining different types of data such as RSSI with magnetic, and Time of Arrival (ToA) [7,8]. In an indoor environment these approaches are affected by multipath propagation and scattering.

An ideal robust feature extraction method for ANN-based fingerprinting should maximize the dissimilarity between classes (RP) while minimizing the difference within classes. There exist many mathematical approaches to perform such tasks, and specifically methods based on scattering transform. Linear Discriminant Analysis (LDA) is among such methods. However, the LDA requires more samples per class to compute the scattering parameters. Also, in most available RSSI datasets, there is almost one sample per reference point (RPs). This lack of samples per RP makes the LDA quite difficult to implement. Moreover, the fact of replacing missing data by the same constant for all APs can generate a problem of collinearity when applying the LDA. The wavelet Scattering transform [9] is a promising approach since it has proved to deal with small deformation in signal processing. The wavelet scattering transform possesses the same properties as the LDA to minimize the difference within class while maximizing dissimilarity across classes.

In this paper, we proposed a neural network based fingerprinting algorithm using wavelet scattering framework to extract the feature data from the fluctuating RSSI. The wavelet scattering framework possesses the same properties used in deep learning to extract reliable features from data. Those properties are multiscale contraction, linearization, and sparse representation. Additionally, the wavelet scattering can extract reliable information at different scaling level of decomposition for different scattering paths. Since, the extracted features are insensitive to translation, rotation, and small deformation, the proposed method will not be affected by the handset orientation. Also, the proposed method can be helpful when there is a lack of data by providing features of the same signal at different scales. Such advantages of the wavelet scattering transform could be useful in dealing with RSSI variation in a fingerprinting algorithm. In our work the extracted features consist of the concatenated modulus coefficients of different scattering paths at different levels. The proposed algorithm was evaluated on real environmental data and it has achieved better results in dealing with the impact of RSSI variation.

The rest of this paper is organized as follows: the next section surveys the works related to indoor fingerprinting localization. Section 3 describes our method and in Section 4 the experimentation results are presented. Section 5 discusses the results, then Section 6 concludes our work.

## 2. Related Works

### 2.1. Indoor Localization

Indoor localization based on radio signal characteristics has often been studied as a replacement of GPS positioning and navigation system for an indoor environment. Those methods include the Angle of Arrival (AoA) and its variant Time of Arrival (ToA) and especially the fingerprinting-based methods [10,11]. However, the AoA and ToA are mainly subject to multipath propagation which is natural for an indoor environment where there are many obstacles capable of scattering the signal. Thus, the fingerprinting method has been widely implemented with various kind of RSSI, magnetic field, accelerometer data, etc. In [12], the authors made use of Bluetooth and RFID RSSI. The combination of these types of signal allowed to improve the localization performance by reducing the influence of the RSSI variation. There are also several works that propose the use of magnetic data for indoor localization based on fingerprinting. In [13] magnetic fingerprints are used to perform localization for a multi-building environment. The magnetic field is measured along axes (x,y,z) and used with an ANN. Such a method allows having good building identification based on magnetic pattern. An indoor localization method based on the magnetic field using deep learning is proposed in [14]. In that model, the authors used light intensity data to refine the result obtained with magnetic field.

However, in an indoor environment where users are moving with their handset, the magnetic field can be significantly perturbated. Furthermore, different approaches have been studied to improve the fingerprinting based on the RSSI data. Such methods include the most advanced machine learning algorithms [15,16]. Yu Zhang et al. [17] introduced a tensor decomposition based model to model the fingerprinting data. In addition, to deal with anomaly reading of the data during the collection. The main aim in this work is to distinguish anomaly reading from correct reading and not to address the RSSI fluctuation. Also, methods exploiting the ANN to mitigate the Wi-Fi RSSI fluctuation have been proposed [9,18,19,20,21].

These methods used DNN autoencoder approach to extract discriminative features from the fluctuating RSSI and applied a DNN classifier for target classifications. Such an approach achieves good results in some cases but it is still affected by the RSSI instability. To date, there is no adequate framework for extracting the reliable features from the unstable RSSI signal. Also, the existing ANN-based fingerprinting methods are not stable to small deformations and they are not rotation invariant.

### 2.2. Wavelet Sattering

Recent decades have seen a growing interest in applying wavelet frameworks to signal processing. These works include wavelet transform, wavelet scattering transform and wavelet signal denoising [22]. Mallat et al. [23,24,25,26] have largely investigated the wavelet scattering transform framework and its properties. They demonstrated how the wavelet scattering transform can extract reliable information at different scales. The Scattering transform was tested on handwriting image data to extract the features where it achieved good performance. Also, it has been proved in the literature that the wavelet scattering coefficients are more informative than a Fourier transform when dealing with short variation signals or small deformation and rotation invariant [27,28]. In [29], discrete wavelet transform is used to compressed the input data for an ANN-based localization method. The use of wavelet transform in that work was to compress the input signal to facilitate its use by the ANN. The method has been evaluated on simulation data and has achieved good results.

## 3. Materials and Methods

The main contribution of our study is to extract reliable features from fluctuating RSSI using one-dimensional wavelet scattering transform.To do that, the raw RSSI data is assimilated to a time series data collected at a fixed frequency. In addition, the wavelet scattering transform of the raw RSSI is computed and then the complex coefficients modulus are the inputs of the DNN classifier as described in Figure 1. A trained model consisting of the best fitting weights is obtained after training the model with the scattering coefficient modulus. This trained model is then used to estimate targets’ location probabilities in relation to the RPs. The prediction probabilities are used to estimate the targets’ position.

### 3.1. Wavelet Scattering Transform

Wavelet techniques are effective tools for good data representations and feature extractions which can be used with most available classification algorithms. The wavelet scattering transform allows us to produce reliable features that are locally stable to small deformations which we can use in conjuction with a deep neural network. To produce a wavelet scattering transform of a time series input signal *X*, three successive main operations are required such as convolution, nonlinearity, and averaging as described in Figure 2. The input signal $X=[x_{1},x_{2},\ldots ,x_{n}]$ is a n*n*-dimensional vector data whose length is the number of APs.

In the rest of this paper, *j* and *J* are integers whin j≤J where *j* is the maximum level of scattering. The wavelet functions used in the scattering transform process are dilated mother wavelets with different scaling levelsa. A well-known example of mother wavelet is Mortlet wavelet noted by ψ as defined in Equation (Equation 1) where *i* is the complex number such as $i^{2}=-1$.
(1)$$\psi \left(\mu \right)=C_{1}\left(e^{i\xi .\mu }-C_{2}\right)e^{-{\left|\mu \right|}^{2}/\left(2\sigma^{2}\right)},$$
where ξ is the frequency σ is a measure of the spread or support, and $C_{1},\;C_{2}$ are constants such as ∫ψdμ=0 and $\int \psi^{2}d\mu =1$.

The dilated mother wavelet is computed by scaling the mother wavelet with a scaling factor of the form $2^{-j}\;j\in N$. Let’s assume Ψ be the mother wavelet to be dilated with a scaling factor $2^{-j}$ with *j* varies from 1 to *J* the maximum scattering level order. Let denote by $\Psi_{2^{j}}$ its dilated form. This dilated wavelet can be expressed as shown in Equation (Equation 2).
(2)$$\Psi_{2^{j}}\left(\mu \right)=2^{-j}\Psi \left(2^{-j}\mu \right).$$

Similar to the dilated wavelet, the low-pass filter or averaging function $\phi_{J}$ for a scattering transform at scale $2^{J}$ is a dilation form of a low-pass filter ϕ. that we defined as $\phi_{J}\left(u\right)=2^{-J}\phi \left(2^{-J}u\right)$. A general representation of the wavelet scattering transform is presented in Equation (Equation 3). We denote by
(3)$$\Psi \left(X\right)=e^{i\eta .X}\phi \left(X\right),$$
where η is the frequency and *i* complex number such as $i^{2}=-1$. The Fourier transform $\hat{\phi }\left(\omega \right)$ of ϕ(x) is a real valued function with primary support in low-frequency centered around ω=0. Therefore, $\Psi_{2^{j}}$ is localized around $2^{-j\eta }$ [23,26].

A wavelet scattering transform for a maximum scale factor of $2^{J}$ begins with convolving the input signal *X* with a low-pass filter (averaging function) $\phi_{J}$ which performs an averaging operation on the raw input data to produce an approximative representation which is called scattering coefficient of order 0 at scale $2^{J}$ that we denote by $S_{J}\left[\right]X$. Let $X=[x_{1},x_{2},\ldots ,x_{n}]$ be the input RSSI for a given RP in the RSSI database. The zero^th^ order scattering coefficient is computed by $S_{J}\left[\right]X\left(\mu \right)=X⋆\phi_{J}\left(u\right)$. In fact, the wavelet scattering transform downsamples the signal with respect to the filter bank length and input signal length. Although the coefficients’ modulus is an approximation of the input. Its representation is smooth and differs from the input data as presented in Figure 3. The zero^th^ order wavelet scattering coefficients’ modulus containS most of the signal energy and are the closest representation of the original signal than other scattering orders. Additionally, the signal energy decreases with the level of scattering. Therefore, the high level of scattering coefficients will have lower energy. A wavelet can be considered as a band-pass filter and its dilated form is a dilated band filter. The scaling function or low-pass filter captures the lower signal details whereas the wavelet Ψ captures high-frequency components, thus higher details of the input signal. So, at each level, the lower details of the signal are extracted. Figure 3 shows the difference between the raw RSSI of a single measurement at an RP and its zeroth-order scalogram.

Figure 3b represents the zeroth-order scattering coefficcients of an RSSI measurement from 77 APs. The result of the scattering transform is a vector of 10 elements.

After computing the zeroth-order of scattering coefficients, the remaining scattering process starts by convolving the input signal with dilated wavelet as shown in Equation (Equation 4) whose modulus is then convolved with the averaging function to produce the first scattering order. To simplify notations, we denote $\lambda =2^{j},\;j\leq J$, and $\psi_{\lambda }\left(u\right)=2^{-j}\psi \left(2^{-j}u\right)$.
(4)$$W_{J}X=\left(X⋆\psi_{\lambda }\right),$$
where *X* is the input signal, and ψ(u) is the mother wavelet (e.g., Morlet). The modulus $|W_{J}X_{n}|$ of this convolution result is computed and an averaging operation is performed on the result using the low-pass filter as $|W_{J}X|⋆\phi_{J}$ to produce the scattering coefficients modulus of first order. The input of the next stage is the modulus of $|W_{J}X|$ and the same operations are repeated. We denote $U\left[\lambda \right]X=|X⋆\psi_{\lambda }|$ the wavelet modulus operator for the sub-band λ. let $\phi_{J}$ denote the low pass averaging filter and $P=(\lambda_{1},\ldots ,\lambda_{m})$ a scattering path with *m* represents the number of sub-band frequencies or which is the length of *P*. In other words *P* the ordered product of non-linear and non-commuting operators. The cascading operation for computing U[P]X is written as below in Equation (Equation 5).
(5)$$U\left[P\right]X=U\left[\lambda_{m}\right]\ldots U\left[\lambda_{1}\right]U\left[\lambda_{1}\right]X=|||X⋆\psi_{\lambda_{1}}|⋆\psi_{\lambda_{2}}\left|\ldots \right|⋆\psi_{\lambda_{m}}|,$$
with U[]X=X for the empty path. The output signal at each stage of the scattering transform is rewritten as a function of the modulus and the lowpass averaging function is described in Equation (Equation 6)
(6)$$S_{J}\left[P\right]X\left(u\right)=U\left[P\right]X⋆\phi_{J}\left(u\right)=\sum_{v,u}U\left[P\right]x\left(v\right)\phi_{J}\left(u-v\right)$$
where $S_{J}\left[P\right]X\left(u\right)$ denotes the scattering coefficients modulus of order or level *m* at scale $2^{J}$, with *m* the length of *P*. To simplify notations, we denote $S_{i}X,\;i=1,2,\ldots ,m$ the scattering coefficients’ modulus of order *i*, $E_{0}X=S_{J}\left[\right]X$.

During the scattering process, the information lost in a stage is recovered in the next stage. As far as the scattering level increases the energy of the scattering coefficients tends to zero. Thus, it does not require to have an infinite level of decomposition as a Fourier transform. To derive discriminative information from the scattering decomposition, one can use one level’s coefficient or combination or concatenation of more than two different levels’ coefficients’ modulus. Assume that we are at 3 levels of decomposition, the feature data can be either $\left\{S_{0}X,\;S_{1}X,\;S_{2}X\right\}$, a subset of this set or a combination of elements from this set. In the following Figure 4 one can choose $SS_{2}\left[\lambda_{1}\right]X$ or a subset of $S_{1}X=\left\{S_{J}\left[\lambda_{1}\right]X,S_{J}\left[\lambda_{2}\right]X,S_{J}\left[\lambda_{3}\right]X,S_{J}\left[\lambda_{4}\right]X\right\}$ or a combination of elements of the second level order coefficients’ modulus $S_{2}$. The choice of scattering for feature representation is made based on experimental results.

The following diagram Figure 4 describes a multiresolution wavelet scattering transform propagator using four paths at each stage where $S_{J}\left[\emptyset \right]X=X⋆\phi_{J}$ the $zero^{th}$ order coefficients’ modulus, $S_{j}\left[\lambda_{1}\right]X=U\left[\lambda_{1}\right]X⋆\phi_{J}=\left|X⋆\Psi_{\lambda_{1}}\right|⋆\phi_{J}$, and $S_{j}\left[\lambda_{1},\lambda_{2}\right]X=U\left[\lambda_{1}\right]U\left[\lambda_{2}\right]X⋆\phi_{J}=||X⋆\Psi_{\lambda_{1}}\left|⋆\Psi_{\lambda_{2}}\right|⋆\phi_{J}$ The other coefficients are computed based on this procedure as described in Equations (Equation 5) and (Equation 6).

Different experiments were conducted, then the choice of the scattering coefficients for each datasets was depending on their performance on that dataset. Therefore, different level of scattering coefficient were used.

### 3.2. Neural Network Architecture

This section describes the Deep Neural Network model (DNN) used to exploit the extracted features. DNN model predicts the probable RP based on the scattering coefficients from different scattering paths. DNN model consists of four hidden layers and a SoftMax output layer Table 1. The output of the Softmax layer constitutes the prediction probabilities.

We assume that there are *N* RPs, $H_{l}$ last hidden layer output, and $W_{L}$ corresponding weights. We denote by $w_{Li}$ the weights used to ouput the $i^{th}$ Rp prediction. The input of the SoftMax Layer is given by ${W_{L}}^{T}H_{L}=\left[w_{Li}h_{Li}\right]\;i=1,\ldots ,N$. let $Q=[q_{1},q_{2},\ldots ,q_{N}]$ denote the output of the Softmax function and $q_{i}$ the $i^{th}$ element of *Q* associated with the ith RP. Then the output of the DNN network is defined by Equation (Equation 7)
(7)$$q_{i}=\frac{e^{h_{Li}w_{Li}}}{\sum_{j=1}^{N}e^{h_{Lj}w_{Lj}}}\;i=1,\ldots ,N.$$
We constructed the DNN classifier based on the parameters mentioned in the following Table 1. In addition, we adopted a multi-labels classification scheme by using a binary cross entropy loss function to train the network.

The location is derived from the SoftMax output by taking the first 3 RPs which are predicted to be the most closer to the target node with prediction probability greater than a threshold. For *m* test points $T_{j}$, j=1,…,m with exact coordinates. We consider only the reference point $RP_{i},\;i\leq 3$ with prediction probably greater than the threshold that we defined in a descending order with coordinates respectively $(x_{i},y_{i}),\;i=1,\ldots ,3$. Assume that thresh denotes the threshold and $\left(\hat{x_{j}},\hat{y_{j}}\right)$ the estimated position of each taget $T_{j}$ which is computed by Equation (Equation 8).
(8)$$\begin{array}{c} \hat{x_{j}}=\frac{\sum_{i}x_{i}.q_{i}}{\sum_{i}q_{i}}\\ \hat{y_{j}}=\frac{\sum_{i}y_{i}.q_{i}}{\sum_{i}q_{i}}, \end{array}$$
where i≤3 and $q_{i}\geq thresh$. This equation means: for each test point *j* find the first three RPs *i* such as $p_{i}\geq thresh$ with strong probability $p_{i}$ and compute the target position based on these RPs coordinates and those probabilities. The localization error err for *m* test points is defined as in Equation (Equation 9).
(9)$$err=\frac{1}{m}\sum_{j=1}^{m}\sqrt{{\left(x_{j}^{\prime }-\hat{x_{j}}\right)}^{2}+{\left(y_{j}^{\prime }-\hat{y_{j}}\right)}^{2}}.$$

## 4. Experimentation Results and Analysis

To evaluate the performance of the proposed method, experiments were carried out on two different datasets. Matlab wavelet scattering toolbox [30] was used to compute the scattering transform. Then google tensorflow [31] was used to build the DNN model. The same DNN architecture is used for all experiments. All datasets used in this paper are Wi-Fi RSSI datasets.

### 4.1. Local Corridor Experiment

The first experimental data was collected at the corridor of Woncheon Hall at floor four of the department of Electrical and Computer Engineering. An area of 50 m × 1.95 m where 21 RPs were defined and 100 samples per RP was collected with a frequency of 12 seconds between samples. The data was collected by the same user with a Samsung Galaxy S8+. We only used the six strongest APs per RP. A total of 36 APs were retained. Matlab wavelet toolbox was used to achieve the wavelet scattering decomposition. Then the zero order coefficients or scalogram coefficients modulus were used for training as well as for testing data where there were only 11 test samples. We chose the zeroth-order scattering coefficients because these coefficients are those which provided good test accuracy. The data has been normalized before feeding into four layers DNN. The results of the corridor experimentation are presented in Table 2. The experimentation results compared to those in [21] show that the proposed method has reduced the positioning error to around 46%.

### 4.2. Experiment with a Publicly Available Dataset

The proposed method has also been evaluated on a publicly available dataset [32]. This dataset contains 25 subsets of datasets (categorized in months) where each subset contains some training sets (15 for the first month and 1 for others) and test sets (5 per month) separately. This database also contains separated subsets of dataset with some test data collected at the same point as the training data. This makes it useful for checking whether the proposed algorithm is stable to RSSI variation. The performance of the proposed algorithm was evaluated in the first month collection where there are 15 training sets (trn01rss to trn15rss) and 5 test sets l (tst01rss to tst05rss) collected on an area of 308.4 m${}^{2}$ with 576 samples per dataset. In these databases, there are 12 samples for each location where measurements were collected. Also, the test sets tst01rss and tst05rss were collected only at the same points with all training data. The training data contains 48 RPs in total and 24 per floor which can be identified by combining the location coordinates and the floor id (x,y,Floor). Three scenarios were defined to evaluate the proposed method in which evaluation metrics were the floor prediction and the location estimation in two-dimensional coordinates. Floor rate corresponds to the floor classification accuracy.

The first option was defined to prove whether our approach reduces the influence of RSSI variation on the localization performance. In this case, we want the algorithm to find the closest RP or perfect matching RP. To achieve that, we used the test set tst01rss whose targets’ position coincide with RPs’ position in the training data. Training set trn01rss were used and only the first scattering scattering coefficients provided the best results. Thus all experiments using this dataset were based on the first level scattering coefficients. The experimental results presented in Table 3 shows that the proposed method outperformed other methods. In this scenario the autoencoder model (SAE + DNN) was the worst localization method. However, the autoencoder model (SAE + DNN) has been tested with the proposed feature extraction method and has achieved similar results as the proposed algorithm. When applying KNN with the proposed feature extraction method the positioning error become 3.04.

In the second scenario, we evaluated the proposed algorithm using the same evaluation approach presented in the database paper [32] but we only focused on the first month data. In this scenario, all test data were used and the model was trained only with the first training dataset. Below Table 4 we present the experimental results of the scenario 2 or option 2. The proposed algorithm achieved around 99.64% on floor classification and the location error was between 4 m and 5 m with minimum error 4.15 m which is higher than the positioning error achieved by KNN as shown in Table 4. In addition, the autoencoder model presented in [9,21] applied to this database merely achieved 60% on floor classification. The same observation has been proved as in the first option. The (SAE + DNN) model achieved the same performance as the proposed model when using the proposed feature extraction method and there is no much change for the KNN. For the KNN implementation only the first three neighbors were used.

To conclude the performance evaluation of the proposed method, another experiment was carried out which used the first training data of the first month and the first test data of the 25th month. This experimentation is the same as the first scenario only that the test data has been changed. The experimental results (Table 5) show that the proposed method achieved the lower positioning error whereas the SAE + DNN model achieved a good result on this test data than the previous test but it still has a positioning error greater than the proposed method. The KNN was significantly affected by the RSSI variation. The minimum error for the proposed method was 4.27 against 5.43 m for the DNN + SAE.

## 5. Discussion

In this study, we proposed a one-dimensional wavelet scattering transform-based feature extraction approach for RSSI-based fingerprinting localization. A deep neural network (DNN) model was adopted to classify RSSI samples by matching the extracted features with one of the RPs. The DNN classifier provides the probabilities that a given sample is close to an RP. To predict the a target node position after classifying its RSSI measurements, a KNN approach strategy is applied using the coordinates of the first three RPs whose probabilities are greater than 0.2. During the evaluation phase, we did perform several experimentations to determine which combination of scattering coefficients is suitable for the datasets used in this paper. The zero^th^ scattering coefficient were the best choice for the corridor data and the first scattering coefficient was a better choice for the public dataset.Experimentation results showed that the proposed method compared with some existing methods used in this paper has improved the DNN-based fingerprinting localization performance. These results demonstrate that the use of scattering transform can reduce the influence of RSSI variation on an RSSI-based fingerprinting method. Therefore, Wavelet scattering transform is still a promising tool for mitigating the impact of RSSI fluctuation. An explanation of the higher positioning error achieved by the proposed algorithm on the second scenario of the second experiment may be due to an overfitting.

Although we cannot claim that the proposed method is the best existing RSS-based fingerprinting localization method, in our opinion, it provides a new approach for indoor fingerprinting algorithm that can be investigated for localization performance improvements. Further investigations can be done to improve the fingerprinting algorithm based on wavelet scattering transform, such as the used of Support Vector Machine with the scattering coefficients, the exploitation of time-frequency information, Shannon entropy, and autoregressive features.

## 6. Conclusions

In this paper, to reduce the impact of RSSI fluctuation that degrades RSS-based indoor localization methods, a wavelet scattering based feature extraction method for a DNN-based fingerprinting localization method was proposed. The small deformations invariant properties of the wavelet scattering transform coefficients were exploited to reduce the impact of the fluctuating RSSI. The proposed model was evaluated on different datasets. In addition, the results demonstrated that it is a promising approach to deal with the instability of the RSSI. In future works, further investigations can be done using other functionalities of the wavelet framework.

## Figures and Tables

**Figure 1 sensors-19-01790-f001:**
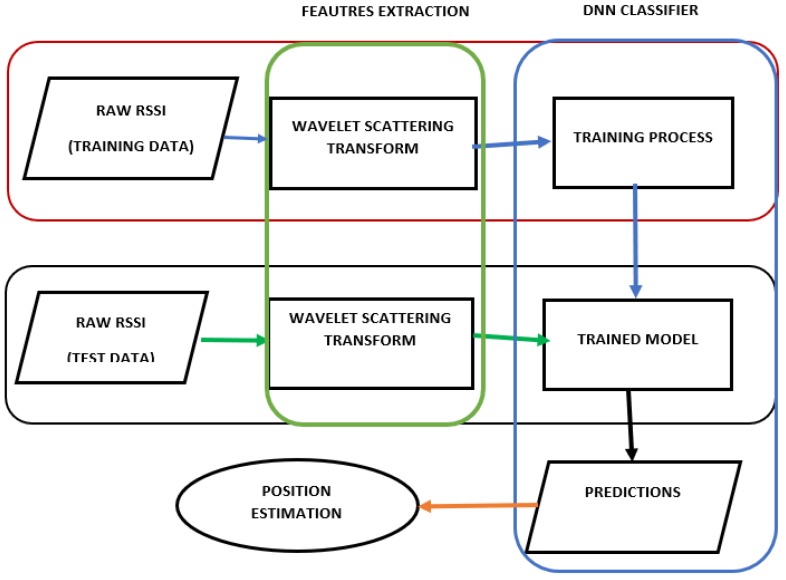
Proposed system architecture.

**Figure 2 sensors-19-01790-f002:**
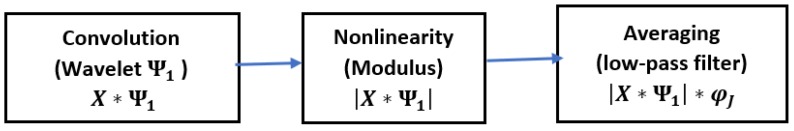
Wavelet scattering transform processes, where *X* is the input data, $\Psi_{1}$ a wavelet function and $\phi_{J}$ an averaging low-pass filter.

**Figure 3 sensors-19-01790-f003:**
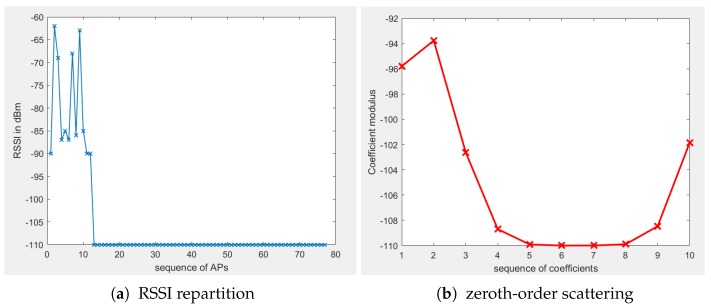
RSSI and its zeroth-order scattering scalogram.

**Figure 4 sensors-19-01790-f004:**
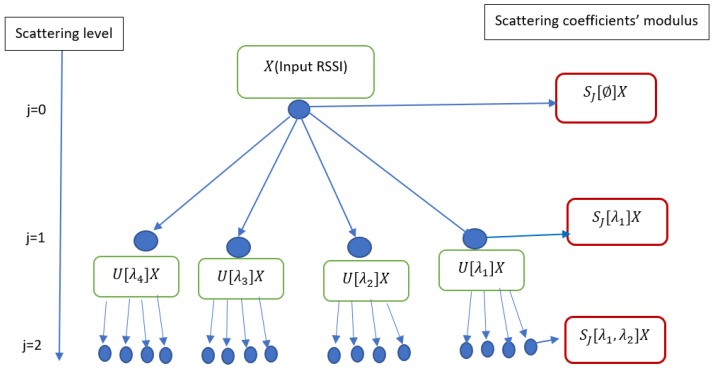
Wavelet scattering decompossition.

**Table 1 sensors-19-01790-t001:** Neural network parameters.

Parametter	Values
Hidden layer	4
hidden neurons	128
hidden activation	relu
output	SoftMax

**Table 2 sensors-19-01790-t002:** Experimentation results on the corridor.

Method	Mean Positioning Error in Meter
SAE + DNN	1.27
MNN	1.58
Proposed	0.68

**Table 3 sensors-19-01790-t003:** Experimentation results for option 1.

Method	Floor Rate (%)	Average Positioning Error (m)
SAE + DNN	50–60	5–17
KNN	100.0	3.07
Proposed	100.0	0.0

**Table 4 sensors-19-01790-t004:** Experimentation results for options 2.

Method	Floor Rate (%)	Average Positioning Error (m)
SAE + DNN	50–60	6–10
KNN	100.0	3.17
Proposed	99.6	4.15

**Table 5 sensors-19-01790-t005:** Experimentation results for option 3.

Method	Floor Rate (%)	Average Positioning Error (m)
SAE + DNN	100.0	5–6
KNN	50.0	7.46
Proposed	95.0	4–5

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
