# Peer review of "A Wavelet Scattering Feature Extraction Approach for Deep Neural Network Based Indoor Fingerprinting Localization†"

_sensors, 2019, doi:10.3390/s19081790_

Round 1

Reviewer 1 Report

The use of wavelets with a Neural Networks was already presented by C. Nerguizian in 2007 in the paper Indoor Fingerprinting Geolocation using Wavelet-Based Features Extracted from the Channel Impulse Response in Conjuction with an Artificial Neural Network. Not significant improvement has been shown in this work.

Author Response

Response to Reviewer 1 Comments

Point1: The use of wavelets with a Neural Networks was already presented by C. Nerguizian in 2007 in the paper Indoor Fingerprinting Geolocation using Wavelet-Based Features Extracted from the Channel Impulse Response in Conjuction with an Artificial Neural Network. Not significant improvement has been shown in this work.

Response to point 1:

C. Nerguizian et al in their paper, have applied a discrete wavelet transform to extract feature of Channel Impulse Response (CIR) information for neural network based fingerprinting localization.  Discrete wavelet transform was applied to compress the input signal in order to reduce the number of neural network's parameters. Their method has been evaluated on simulation data using MATLAB software where they achieve good positioning results than some previously discussed algorithm used in the performance evaluation.

However, in our case, we applied wavelet scattering transform which is a kind of convolutional neural network (Joan Bruna and Stephane Mallat, Invariant Scattering Convolution Networks, https://arxiv.org/abs/1203.1513). In fact, discrete wavelet transform consists of convolving the input signal with a wavelet function whereas the wavelet scattering transform not only computes the same convolutional operation but also performs nonlinearities and averaging operations implemented in the convolutional neural network.

A wavelet scattering transform builds invariant, stable and informative signal representations for classification by scattering the original signal along multiple paths with a cascade of wavelet modulus operators. That makes the scattering transform effective for feature extraction than a discrete wavelet transform.

This work has been cited as reference [29] in our study in section 2.2.

Other major revisions have been done to polish the whole manuscript.

Reviewer 2 Report

This paper proposes a DNN-based indoor fingerprinting localization method, which exploits a wavelet scattering feature extraction approach. Simulation results confirmed its effectiveness and the measured data were obtained from a public database. However, before submitting this manuscript, the authors should extensively polish the whole manuscript. The reviewer has found many typos and mistakes. Some comments were provided as follows:

In abstract, line 5, the full form for the abbr. ``RSSI'' is not ``receiver signal indicator''.

For completeness considerations, please discuss the following related works in Sec. 1.

H. Laitinen et al., "Experimental Evaluation of Location Methods Based on Signal-Strength Measurements," in IEEE Transactions on Vehicular Technology, vol. 56, no. 1, pp. 287-296, Jan. 2007.

P. Pivato et al., "Accuracy of RSS-Based Centroid Localization Algorithms in an Indoor Environment," in IEEE Transactions on Instrumentation and Measurement, vol. 60, no. 10, pp. 3451-3460, Oct. 2011.

S. Fang et al., "Exploiting Sensed Radio Strength and Precipitation for Improved Distance Estimation," in IEEE Sensors Journal, vol. 18, no. 16, pp. 6863-6873, 15 Aug.15, 2018.

Torres-Sospedra et al., ``Off-Line Evaluation of Mobile-Centric Indoor Positioning Systems: The Experiences from the 2017 IPIN Competition'', Sensors, 2018, vol. 18, no. 2, pp. 1 - 27.

 One of your core material in this paper is the wavelet scattering transform. However, in Sec. 3.1, many errors were found in your equation. Many symbols are not well-defined. Please rewrite this section.

 The name of x-axis in Fig. 3(a) is wrong. Also, the unit of RSSI is not ``amplitude''.

The quality of Fig. 4 is poor.

The expression of Eq. 8 is wrong.

The discussion should be improved.

Author Response

Response to Reviewer 2 Comments

Point1: This paper proposes a DNN-based indoor fingerprinting localization method, which exploits a wavelet scattering feature extraction approach. Simulation results confirmed its effectiveness and the measured data were obtained from a public database. However, before submitting this manuscript, the authors should extensively polish the whole manuscript. The reviewer has found many typos and mistakes.

We have carefully reviewed the whole manuscript and major changes have been done in the introduction, methods, and discussion.  Also, we have discussed the references that you have suggested.

In the introduction we have also discussed the reference papers proposed by the reviewer. As the reviewer has mentioned, major revision has been carried out for both method section and discussion. The discussion has been rewritten.

Point 2: In abstract, line 5, the full form for the abbr. ``RSSI'' is not ``receiver signal indicator''.

In the abstract we have corrected the RSSI definition by clearly writing ‘Receiver Signal Strength Indicator).

Point 3: For completeness considerations, please discuss the following related works in Sec. 1.

H. Laitinen et al., "Experimental Evaluation of Location Methods Based on Signal-Strength Measurements," in IEEE Transactions on Vehicular Technology, vol. 56, no. 1, pp. 287-296, Jan. 2007.

P. Pivato et al., "Accuracy of RSS-Based Centroid Localization Algorithms in an Indoor Environment," in IEEE Transactions on Instrumentation and Measurement, vol. 60, no. 10, pp. 3451-3460, Oct. 2011.

S. Fang et al., "Exploiting Sensed Radio Strength and Precipitation for Improved Distance Estimation," in IEEE Sensors Journal, vol. 18, no. 16, pp. 6863-6873, 15 Aug.15, 2018.

Torres-Sospedra et al., ``Off-Line Evaluation of Mobile-Centric Indoor Positioning Systems: The Experiences from the 2017 IPIN Competition'', Sensors, 2018, vol. 18, no. 2, pp. 1 - 27.

H. Laitinen et al., "Experimental Evaluation of Location Methods Based on Signal-Strength Measurements," in IEEE Transactions on Vehicular Technology, vol. 56, no. 1, pp. 287-296, Jan. 2007.

In this paper, the authors investigated the effect of distance, antenna pattern, propagation model, and propagation condition on Receiver Signal Strength-based localization methods for an outdoor environment. Three propagation models were considered: a range dependent model without and with fitting to measurement data, and a site-specific model. those models have been evaluated through different experimentations. However, the methods presented in this study underline outdoor localization models. Since such approaches are based on the path loss model, their application to an indoor environment requires further modifications. Because the indoor propagation model is mostly subject to scattering and fading.

We referred to this work in our investigation to emphasize the important studies that aim to improve RSS based localization performance in reference [3].

P. Pivato et al., "Accuracy of RSS-Based Centroid Localization Algorithms in an Indoor Environment," in IEEE Transactions on Instrumentation and Measurement, vol. 60, no. 10, pp. 3451-3460, Oct. 2011.

In this paper, the authors investigated the weighted centroid RSS-based localization (WCL) and relative-span exponential weighted localization (REWL) methods. those methods are based on distance to transmitter estimation based on the path loss model.

However, such approaches require a wise choice of the weights. Also, there is no typical path loss model that works correctly in an indoor environment.

This work is cited in our study as reference [5] to emphasize studies that have been done to improve the performance of RSS-based localization algorithms.

S. Fang et al., "Exploiting Sensed Radio Strength and Precipitation for Improved Distance Estimation," in IEEE Sensors Journal, vol. 18, no. 16, pp. 6863-6873, 15 Aug.15, 2018.

In this study, the authors proposed an RSS-based localization model that addresses the effect of rain on the positioning performance. They also applied a modified maximum likelihood estimator to estimate target nodes location. However, such a model is likely valid only for outdoor environments where it is subject to rain.

We referred to this study as [4] to emphasize studies in which weather conditions are considered to provide a robust localization model for an outdoor environment.

Torres-Sospedra et al., ``Off-Line Evaluation of Mobile-Centric Indoor Positioning Systems: The Experiences from the 2017 IPIN Competition'', Sensors, 2018, vol. 18, no. 2, pp. 1 - 27.

This work describes the indoor positioning and indoor navigation competition track 3’s results. Although IPIN2017 proposed a rich dataset for indoor localization, it has been difficult for us to understand how to extract the label information for the floor level. Therefore, we could not use this dataset for performance comparison.

We have cited this work in reference [6] to underline the ongoing effort dedicated to the improvement of indoor localization algorithms.

The suggested reference has been included in section 1 (introduction) from line 32 to 43.

We have introduced these studies to underline the fact that some efforts that have been done to propose methods that improve the RSS-based localization algorithms.

 Point 4: One of your core materials in this paper is the wavelet scattering transform. However, in Sec. 3.1, many errors were found in your equation. Many symbols are not well-defined. Please rewrite this section.

The name of x-axis in Fig. 3(a) is wrong. Also, the unit of RSSI is not ``amplitude''.

The quality of Fig. 4 is poor.

The expression of Eq. 8 is wrong.

The discussion should be improved.

We have rewritten section 3.1 as well as most of the equations. Figure 3.a has been replaced with a new one where x-axis (APs) and y-axis(dBm) values have been correctly added. We have replaced figure 4 with another figure without changing the wavelet scattering concept.

After checking the grammar points and wording of section 3.2 and section 4, we have rewritten the discussion section. In the conclusion section, only wording and grammar points were checked.

Also, equation 8 has been clearly redefined

Round 2

Reviewer 1 Report

The paper can be accepted now

Reviewer 2 Report

In this revision, all my previous concerns have been well-addressed. Thus, the reviewer would suggest accepting this paper as the current form.